# Experimentally Induced Dieback Conditions Limit *Phragmites australis* Growth

**DOI:** 10.3390/microorganisms11030639

**Published:** 2023-03-02

**Authors:** Wesley A. Bickford, Danielle S. Snow, McKenzie K. H. Smith, Kathryn L. Kingsley, James F. White, Kurt P. Kowalski

**Affiliations:** 1Great Lakes Science Center, U.S. Geological Survey, 1451 Green Road, Ann Arbor, MI 48105, USA; 2Akima Systems Engineering, Herndon, VA 20171, USA; 3Department of Ecology and Evolutionary Biology, Tulane University, New Orleans, LA 70118, USA; 4Department of Plant Biology, Rutgers University, New Brunswick, NJ 08901, USA

**Keywords:** short-chain fatty acid, soil bacteria, invasive plants, symbiosis

## Abstract

*Phragmites australis* is a cosmopolitan grass species common in wetland ecosystems across the world. In much of North America, the non-native subspecies of *Phragmites* threatens wetland biodiversity, hinders recreation, and is a persistent problem for natural resource managers. In other parts of the world, populations are in decline, as Reed Die-Back Syndrome (RDBS) plagues some *Phragmites* stands in its native range. RDBS is defined by a clumped growth form, stunted root and shoot growth, premature senescence, and shoot death. RDBS has been associated with a build-up of short-chain fatty acids (SCFAs) and altered bacterial and oomycete communities in soils, but the exact causes are unknown. To control invasive *Phragmites* populations, we sought to develop treatments that mimic the conditions of RDBS. We applied various SCFA treatments at various concentrations to mesocosm soils growing either *Phragmites* or native wetland plants. We found that the high-concentration SCFA treatments applied weekly induced strong significant declines in above- and belowground biomass of *Phragmites*. Declines were significant but slightly weaker in native species. In addition, soil bacterial abundance increased, diversity decreased, and bacterial community composition significantly differed following treatments, such that treated pots maintained a higher relative abundance of Pseudomonadaceae and fewer Acidobacteriaceae than untreated pots. Our results suggest that application of SCFAs to *Phragmites* can lead to stunted plants and altered soil bacterial communities similar to populations affected by RDBS. However, the lack of species-specificity and intensive application rate may not make this treatment ideal as a widespread management tool.

## 1. Introduction

*Phragmites australis* is a cosmopolitan grass common in wetland ecosystems across the world and is considered a model species for studying plant invasions due to its global dominance and abundant growth in a variety of ecosystems [1]. In much of North America, non-native invasive *Phragmites* threatens wetland biodiversity, hinders recreation, and is a persistent problem for natural resource managers. Land managers in the United States spend millions of dollars annually fighting the spread of *Phragmites* invasion, often with limited success [2,3]. However, mechanical and chemical management approaches come with significant drawbacks such as off-target effects, limits in duration of impact, and the introduction of novel compounds into natural areas [4,5].

In contrast to the North American invasion, many European *Phragmites* populations are in decline, as Reed Die-Back Syndrome (RDBS) plagues some *Phragmites* stands in its native range. RDBS is defined by a clumped growth form, stunted root and shoot growth, premature senescence, and shoot death [6]. RDBS has impacted *Phragmites* stands in northern, central, and eastern Europe for half a century [7,8] and has more recently been observed in Mediterranean populations as well [9,10,11]. While the exact causes of RDBS are unknown, research indicates that the interaction between eutrophication, litter accumulation, and water table management may play a significant role [8]. Specifically, one hypothesis states that increased productivity driven by eutrophication leads to increases in litter production, and in stagnant waters, the litter accumulates [8]. Short-chain fatty acids (SCFA) such as acetic, butyric and propionic acids and sometimes sulfide are produced during anaerobic decomposition of *Phragmites* litter [8] and have adverse impacts on the health of living plants [6]. As plants succumb to RDBS, these phytotoxins accumulate and further damage *Phragmites* stands [12]. 

The microbial communities in the soils surrounding *Phragmites* populations affected by RDBS may also be implicated in stand health. For instance, the oomycete pathogen, *Pythium phragmitis*, first described in *Phragmites* stands exhibiting signs of RDBS in Germany, has shown high aggressiveness against *Phragmites* leaves and seedlings [13]. In Italian *Phragmites* stands, the oomycete communities were found to be compositionally different, with a significantly higher proportion of *Pythiogeton*, a genus of suspected plant pathogens, in stands showing signs of dieback [9]. Bacterial communities also differed between healthy stands and those exhibiting signs of dieback, with bacterial diversity declining and community structure shifting with increasing stem clumping [10]. Although it is unclear whether microbial community changes associated with RDBS are a cause or effect of dieback, the authors speculate that changes in the bacterial communities associated with RDBS may also allow pathogenic oomycetes to proliferate in the microbiome [10], suggesting cascading effects that may enhance dieback severity over time. 

Worldwide, microbial associations with *Phragmites* tend to be strongly influenced by geography, with some microbes found in *Phragmites* populations across the world [14]. The functions of *Phragmites*-associated microbes range from saprotrophs, to pathogens, to symbionts responsible for disease suppression and nutrient uptake [14,15,16]. In North America, invasion of *Phragmites* does not seem to be driven by microbial associations alone [17,18,19]; however, microbes remain important for key functions (e.g., tolerance to salinity, nutrient uptake) in the *Phragmites* [15,20,21,22], and can enhance invasiveness in some systems [23]. Moreover, invasive *Phragmites’* response to disturbances in the soil microbiota suggests that it may be susceptible to controls targeting the microbial community [19]. 

Invasive *Phragmites* is also thought to acquire some micronutrients (e.g., boron, manganese, copper) from microbes through a process known as rhizophagy, whereby plants extract nutrients from intracellular microbes in root tissues [22]. During the rhizophagy cycle, free living soil microbes obtain nutrients from soils and then enter plant roots through the root tip meristem cells where plant-produced reactive oxygen exposure allows plants to extract nutrients. After nutrient extraction, surviving microbes induce root hair elongation and exit through root hair tips [22]. SCFAs also play a critical role in the rhizophagy cycle; naturally produced SCFAs build up in root biofilms where plant roots can absorb them, triggering infection of roots by biofilm microbes [22]. However, when SCFAs are present in soils in high concentrations, this rhizophagy trigger is halted as biofilm SCFAs are replaced by the surrounding soil matrix suspending microbial entry into roots [22]. Therefore, SCFA addition to soils may harm further plants by interfering with this symbiotic nutrient acquisition. 

Given the extent of *Phragmites* in North America and the resources allocated to control and management [2,3], resource managers are in need of new tools for controlling widespread invasions. A significant effort to design and test new control strategies focusing on the plant–soil–microbial interface has broadened the body of knowledge on the ways that *Phragmites* interacts with microbes and how microbial disruptions could impact *Phragmites’* spread [16]. Here, we sought to develop treatments that mimic the conditions of RDBS by applying various SCFA treatments to mesocosm soils growing either *Phragmites* or native wetland plants. We assessed plant growth and biomass production as well as soil bacterial abundance and community composition following SCFA treatments (calcium butyrate, calcium propionate, butyrate + propionate cocktail) to mesocosm soils. If SCFA treatments are a promising control tool, we would expect to see (1) reduced *Phragmites* productivity or plant death in above- and belowground structures, (2) bacterial community responses that potentially further inhibit growth (e.g., pathogen accumulation), (3) species specificity, where *Phragmites* plants are more strongly impacted by treatments than native species, and (4) scalability, whereby treatments are effective at low concentrations and application intervals.

## 2. Materials and Methods

To test the impacts of various SCFA treatments on *Phragmites* and native plant growth, we designed three randomized experiments (Figure 1). The first tested the impact of acids in low concentration (0.01 M) on *Phragmites* growth using a 4 × 2 factorial arrangement. We varied acid (water control, calcium butyrate, calcium propionate, and calcium butyrate + calcium propionate cocktail) and application rate (weekly or monthly). The second experiment tested the impact of acids in high concentration (0.05 M) on *Phragmites* growth using a 4 × 2 factorial arrangement (4 acids × 2 frequencies). Additionally, to test the impacts of the treatments on multiple plant species, we conducted a third randomized experiment in a 4 × 3 factorial arrangement where we varied acid type (water control, calcium butyrate, calcium propionate, and calcium butyrate + calcium propionate cocktail, all at 0.05 M applied weekly) and plant species (*Phragmites australis*, *Schoenoplectus acutus*, and *Spartina pectinata*). Acid types, combinations, and concentrations were determined based upon the most effective treatments in a preliminary experiment (Appendix A). 

### 2.1. Preparing Mesocosms

In October and November 2017, we collected rhizomes from monoculture patches of *Phragmites australis* (subsp. *australis*) from two locations near Lake Erie’s Ohio coast, Toussaint Creek (41.579143 N, 83.14531 W) and Turtle Creek (41.604008 N, 83.155844 W). Rhizome cuttings were then planted in potting soil in the greenhouse of the Great Lakes Science Center (GLSC) in Ann Arbor, MI, and clones were maintained over multiple years. Material collected from each location was cloned from a single rhizome fragment and was thus considered to be of a single genotype. In March 2019, experimental plants of each genotype were propagated by laying aboveground tillers of mature plants in trays of water. As new shoots emerged from the nodes and developed roots, they were separated and transplanted into potting mix.

To test the impact of the treatments on non-target species, we selected two native wetland plants, *Schoenoplectus acutus* (hardstem bulrush) and *Spartina pectinata* (prairie cordgrass), that share similar growth habits (obvious stems and rhizomatous growth) and environments to *Phragmites*. *S. acutus* and *S. pectinata* plugs were purchased from Wildtype Nursery (Mason, MI, USA) and grown for approximately one month in the GLSC greenhouse prior to planting in experimental mesocosms. Wildtype Nursery states that all seeds (and in some cases cuttings) of the species listed in their catalog were collected in Michigan. Therefore, while the locations and site conditions where seeds were collected are unknown, all plants used in this study (*Schoenoplectus*, *Spartina*, and *Phragmites*) represent genotypes found in Michigan, USA.

We constructed experimental mesocosms from 3.5-gallon buckets by drilling 6 mm holes 15 cm above the bottom edge of the bucket on two opposite sides to facilitate drainage, while allowing for the bottom of the pot to remain moist. Buckets were filled with Garden Magic topsoil (Michigan Peat Co. Houston, TX). One plug of either *Phragmites* (84 total)*, Schoenoplectus* (24 total)*, or Spartina* (24 total) was transferred to each mesocosm. We arranged mesocosms in a randomized block design on the lawn of the Michigan Department of Natural Resources’ Saline Fisheries Research Center (42.155411 N, 83.77526 W). Plants were left to acclimate to the new conditions for one week prior to beginning treatments and remained outdoors for the duration of the experiment.

We constructed 132 mesocosms encompassing three experiments (Figure 1). The first and second experiments were constructed in a 4 × 2 factorial arrangement with 6 replicates resulting in 48 experimental units. Twelve units receiving the “Water Control” treatment were shared between experiments 1 and 2 (Figure 1). The third experiment was designed in a 4 × 3 factorial arrangement with 6 replicates resulting in 72 experimental units. Twenty-four units with *Phragmites* as the focal species were shared between experiments 2 and 3 (Figure 1).

### 2.2. Short Chain Fatty Acid Treatments

Calcium butyrate (C_8_H_14_CaO_4_) and calcium propionate (C_6_H_10_CaO_4_) were tested in this study due to results of preliminary experiments demonstrating inhibition of *Phragmites* growth (Appendix A). Preliminary greenhouse trials suggested that a concentration of approximately 0.05 M was near the acute toxicity limit, as it led to plant death in some instances, whereas concentrations at or above 0.01 M led to minor signs of stress in plants. For this reason, we selected test concentrations of 0.05 and 0.01 M for field testing. In Experiment 1, we prepared 0.01 M solutions using 2.14 g‧L^−1^ calcium butyrate (214.27 g mol^−1^), 1.86 g‧L^−1^ calcium propionate (186.22 g mol^−1^), and (2.14 g calcium butyrate + 1.86 g calcium propionate)‧L^−1^ for the cocktail treatment. In Experiments 2 and 3, we prepared 0.05 M solutions using 10.7 g‧L^−1^ calcium butyrate, 9.3 g‧L^−1^ calcium propionate, and (10.7 g calcium butyrate + 9.3 g calcium propionate)‧L^−1^ for the cocktail treatment.

*Phragmites* mesocosms received weekly or monthly 2 L doses of 0.01 M (Experiment 1) or 0.05 M (Experiments 2 and 3) calcium butyrate, calcium propionate, or calcium butyrate + calcium propionate (cocktail). *Spartina* and *Schoenoplectus* mesocosms (Experiment 3) received weekly 2 L doses of 0.05 M calcium propionate, calcium butyrate, and calcium butyrate + calcium propionate (cocktail). All control mesocosms received 2 L of tap water weekly. All monthly dose mesocosms (Experiments 1 and 2) received 2 L of tap water on the weeks that they were not receiving an SCFA treatment. Treatments were poured directly onto the soil surface.

### 2.3. Monitoring Plant Health

We monitored plants every other week by measuring the tallest living stem to the nearest tenth of a centimeter, counting all other stems and assigning each to 1 of 4 size classes (0–30, 31–60, 61–90, and 90+ cm). *Phragmites and Spartina* stems were measured from the soil surface to the final node. *Schoenoplectus* was measured to the top of the stem, below any emerging inflorescence. We also assigned plants to 1 of 4 stress categories based on yellowing of the leaves and tillers: healthy, mild, moderate, severe. Stress categories were defined by the percentage of the plant tissue showing signs of stress: <15% was considered healthy, 15–45% was mild, 46–65% was moderate, and >65% was severe. If mesocosms contained stolons, they were counted, measured, and reported separately. Broken and/or fully brown stems were noted and counted, as was any evidence of pests. Photos were taken of each plant at every monitoring day against a white backdrop. Dead plants were removed and no longer monitored. Plant death was confirmed by searching near the soil’s surface for actively photosynthetic tissue. If none was found, then the plant was considered dead and harvested. A summary of plant monitoring data is available at [24].

### 2.4. Plant and Soils Harvest and Processing 

All surviving plants were harvested in mid-September after showing signs of senescence. At the time of harvest, plants were removed from the soil and placed into plastic trash bags. For large plants, aboveground material was first cut at the soil level before extracting belowground biomass. The belowground biomass was shaken so that most soil was removed. A bulk soil sample was collected from each mesocosm at the end of the experiment and from one replicate of each treatment midway through the experiment. All plant biomass and bulk soil samples were returned to the lab where plant biomass was stored at 4 °C and bulk soils were transferred to an ultralow freezer at −80 °C until DNA extraction.

Rhizosphere soil was extracted by removing 2–3 roots and suspending them in sterile 50 mL centrifuge tubes with approximately 30 mL of 1X phosphate buffered saline. The tubes were vigorously shaken to loosen soil from roots. The roots were removed using sterile tweezers, photographed, and returned to the rest of the belowground biomass for future processing and data collection. The tubes were then centrifuged at 8000 rpm for 10 min, and the supernatant was poured off down to the 5 mL line. The tubes were vortexed to create a soil/PBS slurry, poured into sterile 15 mL centrifuge tubes, and centrifuged again at 8000 rpm for 10 min. Finally, the supernatant was decanted with a micropipette, and tubes were stored at −80 °C. 

After rhizosphere soil extraction, all belowground tissues were rinsed free of soil. All roots were counted. Roots were examined for evidence of root hair reduction, which could be a sign of interruption of the rhizophagy cycle [22]. Root hairs on primary roots were assessed. Roots were considered to possess or not possess root hairs generally based on whether three 3-inch segments possessed more than five root hairs each. Due to the density of *Schoenoplectus* roots, root counts were estimated in several plants by counting the roots in at least three 3-inch segments and extrapolated to the whole length of the rhizome. Only roots that were attached to the rhizome were counted. Overall root health was characterized by rigidity (rigid or not rigid) and color (white, red, not white, not red). Due to the low number of red roots, we grouped root color as either white or non-white in our analyses. For full color descriptions, see full data release [24]. Aboveground and belowground tissues were placed into separate paper bags and dried at 65 °C until weights stabilized. Dry weights were recorded. A subset of above- and belowground tissues were ground using a benchtop ball mill for tissue nutrient analysis. However, due to issues with contamination and small tissue volumes, nutrient data were unreliable and are not reported here.

### 2.5. Soil Molecular Methods

Soils remained at −80 ℃ for roughly 18 months between collection and extraction due to facility shutdowns during the COVID-19 pandemic. Bulk and rhizosphere soil DNA were extracted from 25 mg (wet weight) of soil using Qiagen DNeasy PowerSoil kits (Qiagen, Hilden, Germany). Manufacturer instructions were followed with two exceptions: (1) soils were vortexed for 12 min, and (2) only 60 mL of Solution C6 was used to elute DNA. DNA quality was checked after each set of extractions with electrophoresis and quantified with a Qubit fluorometer (ThermoFisher Scientific). DNA extracts were stored at −20 ℃. 

Quantitative polymerase chain reactions (qPCR) were performed to quantify relative abundances of bacterial communities in response to treatments. See Appendix A for qPCR conditions and master mixes. Synthetic DNA (IDT gblocks, Integrated DNA Technologies, Coralville, IA) with a 16S rRNA gene sequence covering the V3 and V4 regions was used as the DNA standard for bacteria abundance qPCR (Appendix A).

Data obtained from qPCR were normalized to copies/ng of dry soil; for this conversion, soil moisture content was determined by drying a subsample of approximately 7 g, taken from each sample used for DNA extraction, and dried at 65 ℃ for at least 48 h. Mass was recorded before and after drying to determine soil moisture content. 

The V4 region of the 16S rRNA gene was amplified from bulk and rhizosphere DNA samples using polymerase chain reactions (PCR). Genomic DNA was diluted to ensure equimolar concentration of template DNA in each PCR reaction (2 ng/µL for bacterial PCR). Amplicons were generated using primers described in Kozich et al. (2013), which target the V4 region of the 16S rRNA gene [25]. All PCR reactions were performed using AccuPrime Pfx SuperMix (Invitrogen). See Appendix A for PCR conditions and reagents. Libraries were normalized using SequalPrep Normalization Plate Kit (Life Technologies cat # A10510-01) following the manufacturer’s protocol for sequential elution. The concentration of the pooled samples was determined using Kapa Biosystems Library Quantification kit for Illumina platforms (Kapa Biosystems KK4824). The sizes of the amplicons in the library were determined using the Agilent Bioanalyzer High Sensitivity DNA analysis kit (cat# 5067-4626). The final library consisted of four plates, normalized to the pooled plate at the lowest concentration. Amplicons were sequenced on the Illumina MiSeq platform, using a MiSeq Reagent Kit V2 and 500 cycles (Illumina cat# MS102-2003), according to the manufacturer’s instructions.

### 2.6. Bioinformatics

Raw bacterial sequence data were processed using mothur v1.40.1 [26]. Operational taxonomic units (OTUs) were clustered at 97% for bacterial sequences and assigned to taxonomy by comparing representative sequences to the taxa found in the SILVA database [27]. Bacterial data were rarefied according to the sample that yielded the fewest number of sequences to ensure equal coverage across all samples (11,875 sequences).

### 2.7. Data Analysis

All analyses were carried out separately by experiment in the R environment [28]. Biomass data were square root transformed to conform to a normal distribution. Plant responses among acid treatments (Exp. 1–3), treatment frequencies (Exp. 1–2), and plant species (Exp. 3) were compared using analysis of variance (ANOVA) with a Tukey HSD adjustment. Bacterial alpha diversity was determined by calculating inverse Simpson diversity for each sample and compared among treatments using the same comparisons as with biomass. To explore the differences in community compositions, we removed all single and double-tons and calculated Bray Curtis distances for each sample and compared among treatments using permutational multivariate analyses of variance (PERMANOVA) and homogeneity of dispersions (Perm-DISP) within the vegan package in R. To display differences in community composition, we used principal coordinate analysis (PCoA) plotting the centroids and 95% confidence intervals for each treatment type.

To explore the taxa associated with differences in bacterial community by treatment, we first grouped OTUs by family and then filtered sequence data to include only “abundant” bacterial families, or those that made up 0.01% of the total sequences in a sample (160 of 13,656 OTUs remained in Ex. 3). We then performed multiple Kruskal–Wallace comparisons of family relative abundance by treatment group (and adjusted for multiple comparisons using a Benjamini–Hochberg adjustment) to find families that significantly increased or decreased following treatments. We performed a distance-based redundancy analysis (db-RDA) using backward selection and plotted the results in an ordination with relative abundances of important families included in the model and plotted as vectors. All data and code for analyses can be accessed at [24]).

## 3. Results

### 3.1. Plant Responses

Plants were generally negatively impacted by SCFA applications, but the magnitude of those responses depended on the formulation, concentration, and application frequency. In Experiment 1, the lower concentration applications (0.01 M) tended to reduce *Phragmites* biomass in the weekly dose, but those differences were not significant (ANOVA; acids: F = 1.756, *p* = 0.171; Figure 2a). In contrast, the higher concentration doses (0.05 M) in Experiment 2 were more effective at reducing *Phragmites* biomass (ANOVA; acids: F = 17.029, *p* < 0.001), especially in the more frequent application rate (ANOVA; acid x frequency: F = 4.272, *p* = 0.010). The calcium butyrate + calcium propionate cocktail treatment significantly reduced *Phragmites* biomass relative to control in monthly applications, whereas all acid treatments in the weekly applications significantly reduced *Phragmites* biomass (Figure 2b) and led to premature plant death in most plants (Table 1). Trends were similar in above- and belowground parts, but belowground biomass responded more strongly to treatments than aboveground (Appendix A). The acid treatments also negatively impacted the native plant species. In Experiment 3, all three plant species similarly reduced biomass following organic acid applications (ANOVA; acids: F = 39.239, *p* < 0.001, acid x species: F = 0.879, *p* = 0.516; Figure 2c). However, plant death was more common in *Phragmites* than in native plants (Table 1).

Acid treatments also affected the structure and quality of plant roots, again most prevalent in the high-concentration, high-frequency applications. For instance, following 0.05 M weekly applications, 44% of *Phragmites* plants had loss of root rigidity, and 75% had non-white roots, compared to 0% observed with non-rigid or non-white roots in control treatments (Appendix A). Although acid treatments did not significantly decrease the absolute prevalence of abundant root hairs (ANOVA; acids: F = 1.547, *p* = 0.218; Appendix A), pots receiving weekly acid doses tended to have a lower proportion of roots with abundant root hairs than monthly dosed pots (ANOVA; frequency: F = 3.462, *p* = 0.071; Appendix A).

### 3.2. Bacterial Responses

Soil bacterial richness was higher in bulk (mean = 2,131 OTUs) than in rhizosphere (mean = 2,032 OTUs) soils across all experiments (ANOVA, soil fraction: F = 6.52, *p* = 0.011). However, inverse Simpson diversity did not differ between soil fractions across all experiments (ANOVA, soil fraction: F = 0.71, *p* = 0.40). Additionally, the effects of acid treatments on richness, diversity, and community composition were not different between bulk and rhizosphere soils (i.e., no Acid × Soil Fraction interactions). Therefore, we report only rhizosphere soil analyses as these soils are more likely to impact plant growth due to their proximity to roots.

Soil bacterial diversity was negatively impacted by the acid treatments at the 0.05 M concentration, but not at 0.01 M (Figure 3a,b). Following 0.05 M treatments, inverse Simpson diversity declined by an average of 54% in monthly doses and 74% in weekly doses. Soil inverse Simpson diversity did not differ among host species (ANOVA, host: F = 1.13, *p* = 0.328), with each host realizing an average of 72% decline following acid treatments (Figure 3c). Bacterial abundance in the rhizosphere was unchanged in response to SCFAs at low concentrations (ANOVA, acid: F = 1.73, *p* = 0.176), but marginally increased following high-concentration, high-frequency treatments (Appendix A; ANOVA, acid: F = 2.74, *p* = 0.078) Bacterial community composition followed a similar pattern and was impacted by all acid treatments, but most heavily impacted at the higher-concentration, more frequent treatments. SCFA type and treatment frequency were both significant predictors of variation in microbial community composition in Experiment 1 (PERMANOVA; acid: F = 3.22, r^2^ = 0.1789, *p* = 0.001; frequency: F = 3.36, r^2^ = 0.0622, *p* = 0.001) and Experiment 2 (PERMANOVA; acid: F = 4.91, r^2^ = 0.2612, *p* = 0.001; frequency: F = 3.20, r^2^ = 0.0567, *p* = 0.003). There was also a significant interaction between acid and frequency in Experiment 1 (*p* = 0.039). In Experiment 1 and 2, bacterial communities shift negative along PCo1 with increasing frequency and when acid types are combined (Figure 4a,b).

SCFA type and plant host were significant predictors of variation in bacterial community composition in Experiment 3 (PERMANOVA; acid: F = 8.63, r^2^ = 0.2902, *p* = 0.001; host: F = 2.10, R^2^ = 0.0471, *p* = 0.014). Due to the low amount of variation in bacterial communities explained by plant host species, we combined all host species to understand how the acids affected bacterial communities. With all host species combined, SCFA application shifted communities negative along PCo1. In addition, the application of the cocktail treatment shifted the communities positively along Pco2 (Figure 4c). The biological significance of these community shifts following SCFA treatments is explored in the next section as taxonomic loadings are examined.

After application of SCFAs, several bacterial families increased and some decreased. For instance, in the positive direction of CAP1, which explained 51.96% of the variation in the model, relative abundance of Acidobacteriaceae and Xanthobacteraceae significantly increased while Pseudomonadaceae, Xanthomonadaceae, and Comomonadaceae decreased. These changes in relative abundance represent a depletion of Acidobacteriaceae and Xanthobacteraceae and an enrichment of Pseudomonadaceae, Xanthomonadaceae, and Comomonadaceae in the soils of pots receiving the SCFA treatments (Figure 5). The taxa in Figure 5 represent those present at the end of the experiment. Soil samples collected midway through the experiment suggest that the same trends were emerging, where high-concentration SCFA treatments were enriched in Pseudomonadaceae and were depleted in members of Acidobacteriaceae compared to controls (Appendix A).

## 4. Discussion

SCFA treatments to soils showed modest promise in reducing the growth of *Phragmites australis*. For instance, SCFA treatments significantly decreased above- and belowground plant biomass and increased plant death when acid concentrations were high (0.05 M), application frequency was high (weekly), or both. Additionally, we saw significant enrichment of certain potentially pathogenic bacteria (e.g., members representing Pseudomonadaceae) following SCFA treatments. Finally, although the SCFA treatments negatively impacted all plant species tested, the impacts to *Phragmites* plants were more severe, where biomass declines were more dramatic (Figure 2c) and plant death was more common (Table 1). However, despite promising aspects, scalability of these treatments may be challenging for managers, limiting their broad application as a control measure. For example, SCFA treatments were not effective at low concentrations and were much less effective at less frequent application intervals (Figure 1). Application rates of 0.05M are much higher than what has been documented in field plots experiencing RDBS or in what has been previously tested in laboratory settings. Additionally, manual weekly applications could represent a significant burden for land managers, especially noting the scale of *Phragmites* invasion in North America. Below, we compare the effects of SCFA treatments on *Phragmites* plants and soil microbes to the impacts of RDBS and conclude with implications for management in North America.

Our results are consistent with Armstrong and Armstrong (2001), who showed decreased root growth following application of propionic and butyric acids. However, that study found that propionic acid was toxic at 0.1 M and butyric acid was toxic at roughly 0.001 M. Our results suggest that toxicity is moderated by growth in the mesocosm, as plant death was common in the 0.05 M, weekly dosed pots, but not in the lower concentration or lower-frequency pots. There may be multiple reasons for this discrepancy including buffering capacity or pH of the substrate in our mesocosms [12]. Another factor is the likely difference in residence time of the SCFAs. In natural wetlands experiencing RDBS, SCFA phytotoxins are constantly being produced from rotting plant litter, roots, and rhizomes, which likely keep SCFA concentrations relatively stable. Our mesocosms were treated with SCFAs either weekly or monthly and were then subjected to rainfall and evaporation, which likely makes SCFA concentrations fluctuate over time. Over the study period, rainfall was above average (approximately 2.5 cm above monthly average) [29], which could be responsible for the lower toxicity when compared to values in the literature. Therefore, only the high-concentration, high-frequency application maintained soil concentrations high enough to induce plant death.

The mechanisms driving plant death and biomass declines are unclear, but they could be direct toxicity, disruption of symbioses, increased susceptibility to pathogens, or a combination of each. Loss of root rigidity was exhibited in 44% and discoloration in 75% of the weekly acid-treated pots, compared to 0% in control pots. This could be evidence of direct toxicity whereby roots are losing turgor pressure or evidence of necrosis from pathogenic microbes. SCFA treatments may have also impacted the rhizophagy cycle as evidenced by the trend toward fewer root hairs in acid-treated pots (Appendix A), which has been found in other plant species exposed to SCFAs [30] and could limit bacterial nutrient exchange between root endosphere and soil [22]. Lacking reliable nutrient response data, we are unable to speculate upon the magnitude of impact from rhizophagy disruption.

Changes in bacterial communities resulting from SCFA treatments showed similarities to soil microbes in wetlands experiencing RDBS. For instance, we found distinct bacterial communities in soils resulting from individual SCFA treatments (Figure 4), consistent with observations of bacterial [10] and oomycete communities [9] in soils experiencing RDBS. In our study, at high application frequency and high concentration, the differences in bacterial communities were coupled with a decrease in bacterial diversity (Figure 3) and a modest increase in abundance (Appendix A), which suggests that the SCFA treatments created conditions suitable for a select few taxa to thrive. Our explorations into bacterial families further showed that some families proliferated and others declined following SCFA treatments. For instance, all soils receiving SCFA treatments significantly decreased in abundance of Acidobacteriaceae and Xanthobacteraceae, and those receiving the cocktail treatment significantly increased in Pseudomonadaceae and Xanthomonadaceae. While these results represent the bacterial families surviving to the end of the experiment, samples collected from the middle of the experiment suggest that the same trend was emerging (Appendix A). This bacterial response is similar to that of European *Phragmites* stands experiencing RDBS where some taxa increase in prevalence and others decrease [10].

The high-frequency, high-concentration cocktail treatment was the most effective at reducing *Phragmites* growth and inducing plant death. Whether the increase in Pseudomonadaceae and Xanthomonadaceae relative abundance is a cause or effect of plant stress and death is unresolved. Pseudomonadaceae and Xanthomonadaceae are diverse families that contain, among many others, species known to be plant pathogens [31]; thus, it is conceivable that the increase in abundance could indicate an accumulation of pathogens that are attacking belowground organs and limiting productivity. Additionally, it is possible that the disruption in the bacterial communities we documented allowed members of other microbial groups, such as fungi or oomycetes, to proliferate. A weakness of this study was that we were unable to document the impact of our treatments on oomycetes. In natural stands experiencing RDBS, others have speculated that disruptions in bacterial communities allow oomycete pathogens to proliferate [10], and those increases in oomycete pathogens have been documented in RDBS stands [9] and may contribute to further degradation of plant health.

Another plausible scenario for the shift in bacterial community composition is that the community differences result from the microhabitats created by the acids selecting for particular taxa. For instance, some species of Geobacteraceae, which were prevalent in the Ca-butyrate pots, are capable of fermentative and syntrophic growth [32]; thus, it is possible that they are metabolizing byproducts of other microbes or using organic materials in the acids themselves as electron donors. Finally, if the acids are harming the plants directly, such that root and rhizome necrosis is occurring, the changes in microbial communities could represent a buildup of decomposers helping to break down dead belowground organs.

SCFA treatments did not appear to be entirely species specific, although the mortality rate was higher in *Phragmites* than in native species (Table 1). Most reported stands of *Phragmites* affected by RDBS in Europe are monocultures; thus, it is unclear if other species are as heavily impacted in field settings. However, reported impacts of RDBS include loss of shoreline stabilization, increased sediment transport, and increases in open water [33], suggesting that little or undesirable plant cover follows RDBS. In a management context, species specificity is desirable but not required. Most *Phragmites* control treatments commonly used are not species specific [3]. The propensity for *Phragmites* to create dense monocultures often means that few desirable plant species are present in treated areas, thus reducing the need for species specificity of control treatments.

Despite the promising results observed in mesocosm trials, inducing die-back conditions likely has limited viability as a control tool due to practical considerations surrounding scalability. For instance, SCFA treatments were only effective when applied at the highest concentration and frequency combination (Figure 2). A management tool that requires weekly applications in high concentrations is likely not preferable for many land managers, especially given the scope of the problem and current investments in control [2,3]. Due to its multivariate causes, sustaining conditions of RDBS by manual SCFA application is difficult to achieve artificially and may not be desirable in all regions. Therefore, sustaining widespread dieback conditions as a form of control may not be a viable strategy for controlling *Phragmites* in many parts of the United States.

## 5. Conclusions

We showed evidence that inducing dieback conditions through additions of SCFAs to soils can significantly decrease above- and belowground plant biomass and increase *Phragmites* death more than in native plants when acid concentrations were high (0.05 M), application frequency was high (weekly), or both. Mechanisms behind biomass reductions were unclear but could result from either direct toxicity, disruption of symbioses, increased susceptibility to pathogens, or a combination of each. Despite promising aspects of these treatments, scalability may be challenging for managers limiting their broad application as a control measure. Although direct SCFA application may not be a viable control alternative, this research provides a foundation for a promising direction for future research into direct and indirect control of microbial interactions that may influence *Phragmites* growth and survival.

## Figures and Tables

**Figure 1 microorganisms-11-00639-f001:**
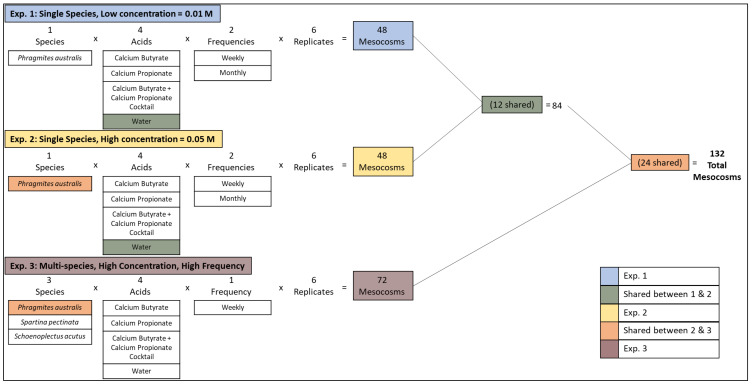
Schematic showing experimental design of each experiment in the full study. Some mesocosms were shared among multiple experiments. Colors indicate which experiment(s) each mesocosm was used in.

**Figure 2 microorganisms-11-00639-f002:**
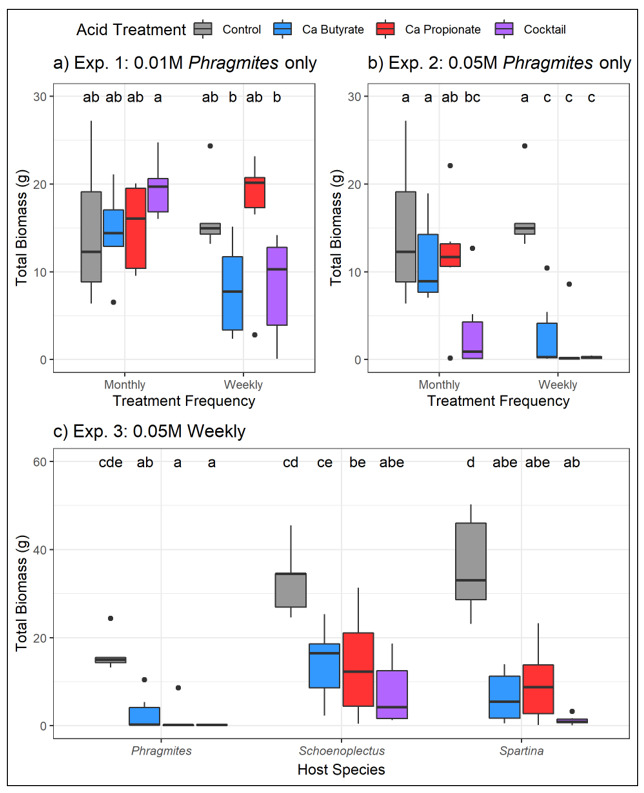
Total plant biomass response to SCFA treatments in (**a**) low-concentration *Phragmites* mesocosms, (**b**) high-concentration *Phragmites* mesocosms, and (**c**) high-concentration mesocosms of *Phragmites* or native species. Colors indicate SCFA type. Letters indicate significant differences among all treatments within an experiment following ANOVA with post hoc Tukey HSD adjustment.

**Figure 3 microorganisms-11-00639-f003:**
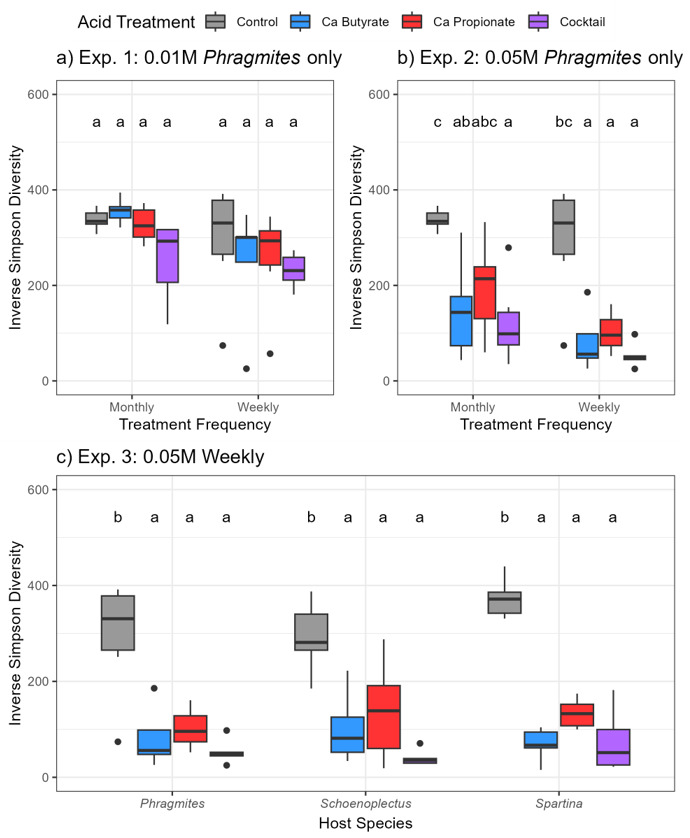
Inverse Simpson diversity of soil bacteria following SCFA treatments in (**a**) low-concentration *Phragmites* mesocosms, (**b**) high-concentration *Phragmites* mesocosms, and (**c**) high-concentration mesocosms of *Phragmites* or native species. Colors indicate SCFA type. Letters indicate significant differences among all treatments within an experiment following ANOVA with post hoc Tukey HSD adjustment.

**Figure 4 microorganisms-11-00639-f004:**
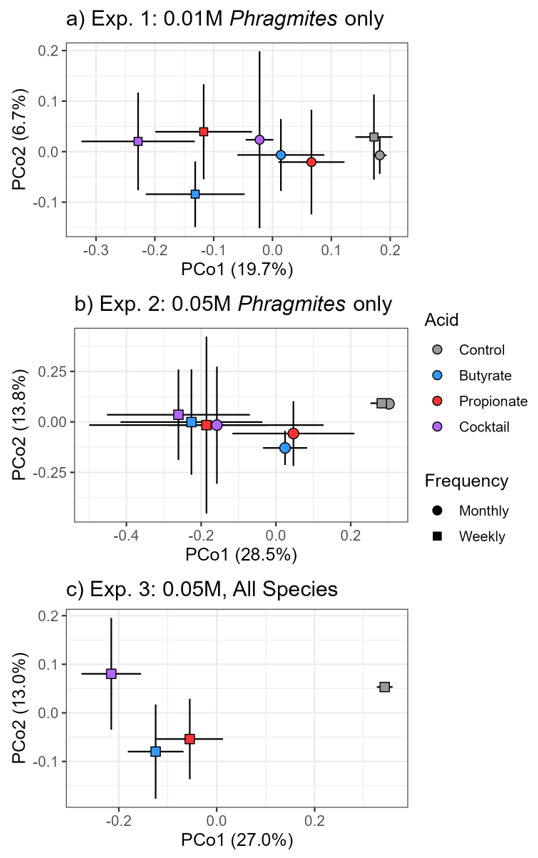
Ordination plots of principle coordinate analysis (PCoA) indicating bacterial community differences among SCFA type and frequency in (**a**) low-concentration *Phragmites* mesocosms, (**b**) high-concentration *Phragmites* mesocosms, and (**c**) high-concentration mesocosms of *Phragmites* or native species. Colors indicate SCFA type. Points represent centroids of all mesocosms within each treatment type. Bars represent 95% confidence intervals around centroids.

**Figure 5 microorganisms-11-00639-f005:**
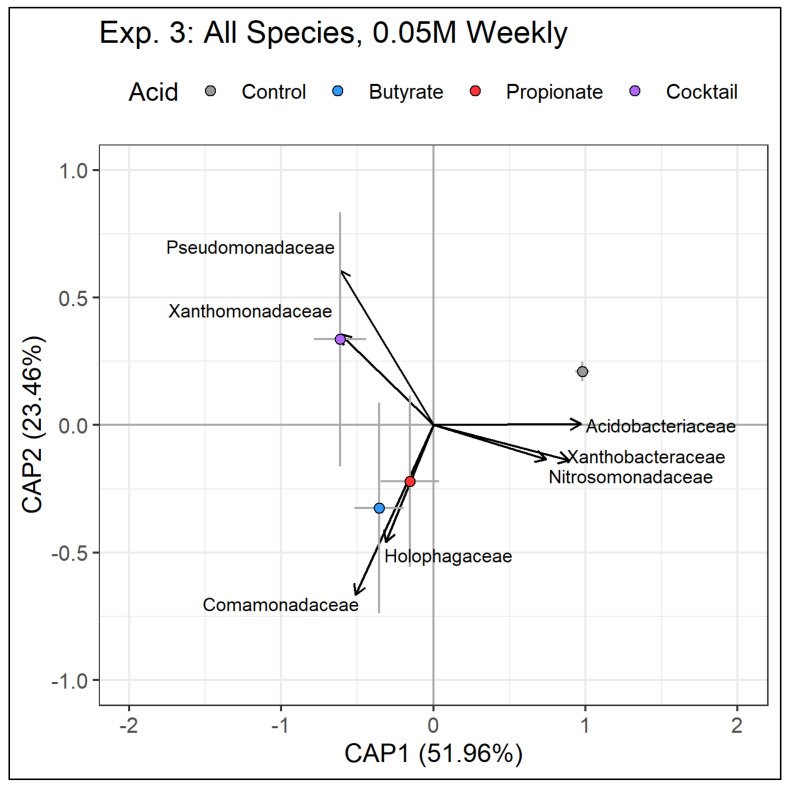
Ordination representing output from a distance-based redundancy analysis (db-RDA) of bacterial communities present in mesocosms included in Experiment 3. Bacterial families that were significant predictors of variation were included via backward selection. Points represent centroids of all mesocosms within each treatment type. Bars represent 95% confidence intervals around centroids. Vectors indicate relative abundance loadings of significant bacterial families.

**Table 1 microorganisms-11-00639-t001:** Proportion of mesocosms that were completely dead following SCFA treatments in (a) low-concentration *Phragmites* mesocosms, (b) high-concentration *Phragmites* mesocosms, and (c) high-concentration mesocosms of *Phragmites* or native species.

(a) Experiment 1: 0.01 M			
Acid Treatment	Proportion Dead Monthly Doses	Proportion Dead Weekly Doses	
Control	0.0	0.0	
Ca Butyrate	0.0	0.167	
Ca Propionate	0.0	0.0	
Cocktail	0.167	0.167	
**(b) Experiment 2: 0.05 M**			
Acid Treatment	Proportion Dead Monthly Doses	Proportion Dead Weekly Doses	
Control	0.0	0.0	
Ca Butyrate	0.0	0.667	
Ca Propionate	0.167	0.833	
Cocktail	0.5	0.833	
**(c) Experiment 3: 0.05 M**			
Acid Treatment	Proportion Dead *Phragmites*	Proportion Dead *Schoenoplectus*	Proportion Dead *Spartina*
Control	0.0	0.0	0.0
Ca Butyrate	0.667	0.0	0.167
Ca Propionate	0.833	0.167	0.167
Cocktail	0.833	0.5	0.667

## Data Availability

All data and codes for analyses can be accessed at [24].

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
