# Peer review of "Experimentally Induced Dieback Conditions Limit Phragmites australis Growth"

_microorganisms, 2023, doi:10.3390/microorganisms11030639_

Round 1

Reviewer 1 Report

Dear editor,

I reviewed the manuscript entitled Experimentally induced dieback conditions limit Phragmites growth by Bickford et al.  I found the manuscript relevant to the scope of the journal, the topic is of wide interest for the readers. Authors tried to demonstrate if SCFA application at various concentrations could limit P. australis development. They found few encouraging results, but the mechanisms behind are still unclear.

The authors only considered SCFA treatments to mimic the RDBS syndrome, this can be acceptable, but can the authors say something about the interaction of SCFAs with the bacteria and oomycete mentioned?

I recommend inserting the species name “australis” in the tile.

Introduction

The introduction section can be improved by explaining the connection between SCFA and microorganisms.

Line 53-54: Please, insert the reference of the hypothesis.

Did the authors checked also the reproductive strategy adopted by P. australis when stressed or not stressed? It is observed that sexual reproduction in terms of seed production and germination is enhanced when stress occurs.

Materials and Methods - Results

I find M&M section well structured, with previous test reported in Supplementary materials, an adequate number of replication and controls. Below some comments:

Which P. australis genotype was used in this study?

I would suggest dividing paragraph 2.3 into two, one about the treatment and one about the monitoring of plants growth.

Line 297: dose

Did the authors check for root biomass and aboveground biomass separately? It could be interesting to see the data.

Also seeing the pictures of roots after the treatments is recommended.

Line 305-308: does this statement refer to fig 2c? Please clarify.

Results in line 318-326 are not supported by any figure or data available.

Discussion

I would avoid citing figures again in the discussion section.

Author Response

We thank the editor and both reviewers for their thoughtful comments, concerns, and suggestions about our paper. We have worked to address all their points below. We also received an internal review of our manuscript which provided helpful comments. Those are also incorporated into the most recent draft.

Reviewer 1:

Dear editor,

I reviewed the manuscript entitled Experimentally induced dieback conditions limit Phragmites growth by Bickford et al.  I found the manuscript relevant to the scope of the journal, the topic is of wide interest for the readers. Authors tried to demonstrate if SCFA application at various concentrations could limit P. australis development. They found few encouraging results, but the mechanisms behind are still unclear.

The authors only considered SCFA treatments to mimic the RDBS syndrome, this can be acceptable, but can the authors say something about the interaction of SCFAs with the bacteria and oomycete mentioned?

I recommend inserting the species name “australis” in the tile.

Response: Revised.

Introduction

The introduction section can be improved by explaining the connection between SCFA and microorganisms.

Response: The connection between RDBS and microbes as documented in the literature is explained in the paragraph from lines 62-74. Additionally, we introduce another connections between SCFA build-up and microbes in the paragraph from lines 84-96 which relates directly to bacteria and a potential mechanism underlying our results in this study.  We feel that this provides a broad overview of the known connections between microbes and RDBS.

Line 53-54: Please, insert the reference of the hypothesis.

Response: Revised. The citation in the next sentence applies to this as well. We added it to both to be clear

Did the authors checked also the reproductive strategy adopted by P. australis when stressed or not stressed? It is observed that sexual reproduction in terms of seed production and germination is enhanced when stress occurs.

Response: Unfortunately, that was beyond the scope of this study. The plants in our experimental mesocosms did not go to seed. In our experience, it is rare that potted Phragmites plants produce seeds, especially after only one season.

Materials and Methods - Results

I find M&M section well structured, with previous test reported in Supplementary materials, an adequate number of replication and controls. Below some comments:

Which P. australis genotype was used in this study?

Response: We used the invasive genotype for this study. Added subsp. australis to line 136

I would suggest dividing paragraph 2.3 into two, one about the treatment and one about the monitoring of plants growth.

Response: We agree that the section titles should be modified. We changed the section title in Line 167 to “Short Chain Fatty Acid Treatments” and moved the next section title to line 193 and called it “Monitoring Plant Health”

Line 297: dose

Response: Revised

Did the authors check for root biomass and aboveground biomass separately? It could be interesting to see the data.

Response: Yes, above and belowground biomass was separated. Now reported separately in the supplement (Fig S1&2) and mentioned in the results.

Also seeing the pictures of roots after the treatments is recommended.

Response: That is a good suggestion. We do not think they are appropriate for the main text, however. We do have many root pictures and can provide a few for comparison in the supplement if the reviewers and editor agree they are important.

Line 305-308: does this statement refer to fig 2c? Please clarify.

Response. The first part of the sentence refers to Fig 2c, but the second part (about death) refers to the table. Added a reference to the figure parenthetically.

Results in line 318-326 are not supported by any figure or data available.

Response: Added Root hair figures to the supplement.

Discussion

I would avoid citing figures again in the discussion section.

Response: Thanks for this comment. We think that referencing figures in the discussion helps the reader connect our interpretations to the data presented in our study. We would prefer to keep these references unless the reviewers or editor feel very strongly about it.

Reviewer 2 Report

This study examines the effects of applied short chain fatty acids (SCFAs) to simulate Phragmites dieback in field mesocosms. The Abstract and Introduction state that dieback is also associated with increases in oomycetes (Pythium) and shifts in bacterial communities. While bacteria are assessed using qPCR and shifts in bacterial community composition are reported, there is no mention of oomycetes in Methods or Results. Do the methods used to detect bacteria also detect oomycetes? The researchers should address why they did not report oomycetes, and to report on oomycetes if possible. From my reading of the Introduction, Pythium is considered to be a major pathogen, perhaps more so than potential pathogenic bacteria detected, so its omission in the Results is a major weakness of the study.

Another point I am curious about is the concentrations of SCFAs in soils affected and unaffected by Phragmites dieback. Have SCFAs been measured from soil extracts? It would be of interest to know how the SCFA application rate compares with concentrations in soils.

Overall the experiment is well designed. Other than my two major points about detecting oomycetes and field concentrations of SCFAs, I have a few questions to clarify Methods and Results. It is of course disappointing that this study did not provide a new approach to controlling invasive Phragmites, but the researchers have made the case that this is a first step to understanding processes involved in natural dieback.

Methods

Fig. 1 What are concentrations of SCFAs in field conditions?

130 Were Phragmites populations used in the experiment native or invasive genotypes? Would the experimental results be different if native vs. invasive Phragmites had been tested?

142 “Schoenoplectus acutus (prairie cordgrass) and Spartina pectinata (hardstem bulrush)”

 common names are reversed with Latin names (Schoenoplectus is bulrush, Spartina is cordgrass)

144 What is the provenance of the nursery-grown cordgrass and bulrush plants? Are they from similar environmental conditions as Phragmites populations used?

2.6 Bioinformatics—were oomycetes detected?

297 “does” = doses

320 “75% had non-white roots”  

Methods say red and non-red roots were also assessed. Does this refer to Phragmites roots, or only other species? Are there any results to report for red roots?

323 “pots receiving weekly acid doses tended to have a lower proportion of roots with abundant root hairs than monthly dosed pots”

Report the proportion values (in text would be appropriate, as I believe there are only 4 mean values to report)

370 “application of the cocktail treatment shifted the communities positively along PCo2 (Fig. 4c).“

What does a positive shift mean in biological terms?

413 mesocosms were subjected to rainfall—state how much rainfall occurred during the experimental period.

453 “changes in microbial communities could represent a buildup of decomposers”

Was there a buildup of decomposers? Or is this a hypothesis?

Table S1 what concentration of SCFAs was used for these experiments?

Author Response

We thank the editor and both reviewers for their thoughtful comments, concerns, and suggestions about our paper. We have worked to address all their points below. We also received an internal review of our manuscript which provided helpful comments. Those are also incorporated into the most recent draft.

Reviewer 2:

This study examines the effects of applied short chain fatty acids (SCFAs) to simulate Phragmites dieback in field mesocosms. The Abstract and Introduction state that dieback is also associated with increases in oomycetes (Pythium) and shifts in bacterial communities. While bacteria are assessed using qPCR and shifts in bacterial community composition are reported, there is no mention of oomycetes in Methods or Results. Do the methods used to detect bacteria also detect oomycetes? The researchers should address why they did not report oomycetes, and to report on oomycetes if possible.

Response: Unfortunately, we only examined the effects of our treatments on bacterial communities in the soils, not oomycetes. We address the omission of oomycetes in lines 551-558 and also changed the wording in multiple locations from “microbial communities” to “bacterial communities” to make it clear that our study was limited to bacteria.

From my reading of the Introduction, Pythium is considered to be a major pathogen, perhaps more so than potential pathogenic bacteria detected, so its omission in the Results is a major weakness of the study.

Response: Yes, unfortunately this was beyond the scope of our study to also examine oomycete communities. We agree that understanding how oomycetes react to these treatments would strengthen the paper and have added content to the discussion in lines 551-558to point out that this aspect of the microbiome was not explored in this study.

Another point I am curious about is the concentrations of SCFAs in soils affected and unaffected by Phragmites dieback. Have SCFAs been measured from soil extracts? It would be of interest to know how the SCFA application rate compares with concentrations in soils.

Response: Unfortunately, we did not measure the concentration in the soils, only the rate of application. Our reasoning was, thinking from a management perspective, those applying the treatments would be most concerned about the application rates and the actual concentrations in the soils, while important, would likely be site- and season-specific. So if we found an application rate that was broadly effective, in theory it should work in a broad range of conditions.

Overall the experiment is well designed. Other than my two major points about detecting oomycetes and field concentrations of SCFAs, I have a few questions to clarify Methods and Results. It is of course disappointing that this study did not provide a new approach to controlling invasive Phragmites, but the researchers have made the case that this is a first step to understanding processes involved in natural dieback.

Response: Thank you for your thoughtful insights and fair assessment of our work.

Methods

Fig. 1 What are concentrations of SCFAs in field conditions?

Response: See comment above. We did not measure concentrations in the mesocosms after treatment

130 Were Phragmites populations used in the experiment native or invasive genotypes? Would the experimental results be different if native vs. invasive Phragmites had been tested?

Response: The Phragmites plants used in our experiment we all the invasive genotype. We added (subsp. australis) to line 136. As to whether the native genotype would have responded differently, we did not test that directly. However, given that the other native species were also negatively impacted and the bacterial community responses were not different due to host genotype, we could speculate that they would also be negatively impacted by the treatments. The magnitude of that response is unknown though.

142 “Schoenoplectus acutus (prairie cordgrass) and Spartina pectinata (hardstem bulrush)common names are reversed with Latin names (Schoenoplectus is bulrush, Spartina is cordgrass)

Response: Revised. Thank you for pointing out that mix up and oversight!

144 What is the provenance of the nursery-grown cordgrass and bulrush plants? Are they from similar environmental conditions as Phragmites populations used?

Response: The native plants were grown in a greenhouse at a native plant nursery (Wildtype Native Plants, https://www.wildtypeplants.com/). The environmental conditions at the sites where the seeds were collected is unknown. However, the company is local and states that all seeds (and in some cases cuttings) of the species listed in their catalog were collected in Michigan, USA. This means that both the nursery plants and our Phragmites plants are from wetlands in Michigan. We added this context in lines 148-152.

2.6 Bioinformatics—were oomycetes detected?

Response: See previous comments on this topic. We only explored the bacterial communities, not oomycetes.

297 “does” = doses

Response: Revised 

320 “75% had non-white roots”  

Methods say red and non-red roots were also assessed. Does this refer to Phragmites roots, or only other species? Are there any results to report for red roots?

Response: Thanks for pointing this out. Due to the lack of red roots in any samples, we reported only white vs. non-white roots. I sentence to the methods in line 232-235 to that effect and also placed the root heath data into the supplement.

323 “pots receiving weekly acid doses tended to have a lower proportion of roots with abundant root hairs than monthly dosed pots”

Report the proportion values (in text would be appropriate, as I believe there are only 4 mean values to report)

Response: Thanks for pointing out this omission as well. I have placed a graph of the root hair data into the supplement as well (Fig. S3).

370 “application of the cocktail treatment shifted the communities positively along PCo2 (Fig. 4c).“

What does a positive shift mean in biological terms?

Response: These shifts are likely driven by the winner/loser taxa in response to the SCFA treatments. I added a sentence in lines 390-392 to say the biological significance is addressed in the next figure.

413 mesocosms were subjected to rainfall—state how much rainfall occurred during the experimental period.

Response: Good suggestion. Rainfall was above average during the months the experiment was run. I added a sentence to explain in the discussion in lines 509-511

453 “changes in microbial communities could represent a buildup of decomposers”

Was there a buildup of decomposers? Or is this a hypothesis?

Response: This is a hypothesis. We were not able to analyze function in our study.

Table S1 what concentration of SCFAs was used for these experiments?

Response: There were all at 0.05 M. I added that detail to the figure caption.

Round 2

Reviewer 2 Report

The authors have addressed my concerns and  made appropriate revisions to the manuscript. It is still unfortunate that they do not have sequencing data to detect the presence of oomycetes, but they do acknowledge the lack of these data in the Discussion. Perhaps the researchers will be able to include oomycetes in a future study.